# METΔ14 promotes a ligand-dependent, AKT-driven invasive growth

Marina Cerqua[1], Orsola Botti[1], Maddalena Arigoni[2], Noemi Gioelli[3,4], Guido Serini[3,4], Raffaele Calogero[2], Carla Boccaccio[5,6], Paolo M Comoglio[1,*], Dogus M Altintas[1,*]

*MET* is an oncogene encoding the tyrosine kinase receptor for hepatocyte growth factor (HGF). Upon ligand binding, MET activates multiple signal transducers, including PI3K/AKT, STAT3, and MAPK. When mutated or amplified, *MET* becomes a "driver" for the onset and progression of cancer. The most frequent mutations in the *MET* gene affect the splicing sites of exon 14, leading to the deletion of the receptor's juxtamembrane domain (METΔ14). It is currently believed that, as in gene amplification, METΔ14 kinase is constitutively active. Our analysis of MET in carcinoma cell lines showed that METΔ14 strictly depends on HGF for kinase activation. Compared with wt MET, Δ14 is sensitive to lower HGF concentrations, with more sustained kinase response. Using three different models, we have demonstrated that METΔ14 activation leads to robust phosphorylation of AKT, leading to a distinctive transcriptomic signature. Functional studies revealed that Δ14 activation is predominantly responsible for enhanced protection from apoptosis and cellular migration. Thus, the unique HGF-dependent Δ14 oncogenic activity suggests consideration of HGF in the tumour microenvironment to select patients for clinical trials.

## Introduction

*MET* is an oncogene encoding the tyrosine kinase receptor for hepatocyte growth factor (HGF) (Galimi et al, 1993). Upon ligand binding, MET recruits multiple signal transducers, including PI3K/AKT (i.e., migration), STAT3 (i.e., differentiation), and MAPK (i.e., proliferation) (Ponzetto et al, 1993, 1994; Boccaccio et al, 1998). Upon gene amplification and/or mutations *MET* is altered in multiple cancer types. Exacerbated MET activity confers a functional advantage to cancer cells, unleashing invasive growth (de Luca et al,

1999; Boccaccio & Comoglio, 2006). Surprisingly, the most common mutations observed in cancer patients are not activating mutations of the kinase domain but mutations affecting the splice sites of exon 14. The latter result in exon 14 "skipping" from mRNA with complete deletion of the protein's juxtamembrane domain (METΔ14) (Reungwetwattana & Ou, 2015). It was proposed that the juxtamembrane domain promotes MET degradation, acting as a negative regulator for MET activation (Lee & Yamada, 1994; Villa-Moruzzi et al, 1998; Peschard et al, 2001; Lee et al, 2006; Tan et al, 2010). Therefore, as for *MET* gene amplification, it was suggested that exon 14 "skipping" induces an uncontrolled, autonomous activation of MET and invasive growth. However, drugs targeting MET in patients in clinical trials yielded positive but puzzling results (Paik et al, 2015, 2020; Engstrom et al, 2017; Klempner et al, 2017; Hu et al, 2018; Landi et al, 2019; Moro-Sibilot et al, 2019; Drilon et al, 2020; Wolf et al, 2020). Specifically, only half of the patients harbouring METΔ14 benefited from MET-targeted therapies, suggesting that there are critical aspects of METΔ14 that remained unaccounted for Salgia et al (2020) and Fujino et al (2021).

Here we show that deletion of exon 14 does not result in constitutive activation of the kinase as *MET* amplification/activating mutations do. Upon HGF binding, the pathways triggered by METΔ14 and the transcriptional response are different from those elicited by ligand stimulation or amplification of wt MET. Hence, METΔ14 acts as an HGF-dependent gain-of-function mutation driving a robust and selective AKT activation, rendering cancer cells more prone to survival and migration. These observations may have important mechanistic implications for cancer with METΔ14 mutation.

## Results and Discussion

The *MET* oncogene is mutated in multiple cancer types, highlighting its critical role in cancer progression and invasion (for review, see Graveel et al [2013] and Comoglio et al [2018]). High throughput

[1]Istituto Fondazione di Oncologia Molecolare - La Fondazione Italiana per la Ricerca sul Cancro (IFOM - FIRC) Institute of Molecular Oncology, Milano, Italy  [2]Department of Molecular Biotechnology and Health Sciences, University of Torino, Torino, Italy  [3]Candiolo Cancer Institute-Fondazione del Piemonte per l'Oncologia, Istituto di Ricovero e Cura a Carattere Scientifico, Candiolo, Italy  [4]Department of Oncology, University of Torino School of Medicine, Turin, Italy  [5]Laboratory of Cancer Stem Cell Research, Candiolo Cancer Institute, Fondazione Piemontese per Oncologia - Istituti di Ricovero e Cura a Carattere Scientifico (FPO-IRCCS), Turin, Italy  [6]Department of Oncology, University of Turin Medical School, Turin, Italy

Correspondence: pcomoglio@gmail.com
*Paolo M Comoglio and Dogus M Altintas contributed equally to this work.

sequencing technologies have enabled the discovery of hotspot mutations in *MET,* opening new avenues for next-generation targeted therapies (Comoglio et al, 2018; Guo et al, 2020). We have used the publicly available MSK-IMPACT dataset (Zehir et al, 2017) and found that the splicing site mutations flanking exon 14 of the *MET* gene are by far the most frequent mutations observed in cancer patients (Fig 1A). Uncommonly for a receptor tyrosine kinase (e.g., EGFR, BCL-ABL, FGFR, and SRC in cancer, for review see Du and Lovly [2018]), very few patients displayed mutations in the kinase domain or regulatory regions (S985, Y1003, Y1234, Y1235, etc.). In line with these observations, studies targeting patients with METΔ14 are in progress (Paik et al, 2015, 2020; Engstrom et al, 2017; Klempner et al, 2017; Hu et al, 2018; Landi et al, 2019; Moro-Sibilot et al, 2019; Drilon et al, 2020; Wolf et al, 2020), but the clinical response was curiously restricted to a fraction of patients displaying Δ14 phenotypes (Salgia et al, 2020). These observations reflect the critical importance of the juxtamembrane domain encoded by exon 14 and urge us to explore METΔ14-driven oncogenicity. Mirroring the complexity of cancer multiple alterations may coexist, including *MET* gene amplification in addition to exon 14 "skipping" (Fig 1B). Thus, the common belief suggesting that METΔ14 is constitutively active may be due to the confounding effect of *MET* gene amplification or other activating mutations (Cortot et al, 2017; Descarpentries et al, 2018).

To challenge the postulate of METΔ14 being constitutively active–as opposed to *MET* amplification or established activating mutations–we have tested MET tyrosine phosphorylation, on steady-state, in NCI-H596 cells (lung adenosquamous carcinoma cells, expressing METΔ14 [Ma et al, 2003]), EKVX cells (lung adenocarcinoma cells, expressing wt MET), HS746T (gastric adenocarcinoma cells, METΔ14 amplification), and MKN45 (gastric adenocarcinoma cells, wt MET amplification). For sake of clarity, the cells are therein denoted, respectively, as Δ14, WT, Δ14_Amp, and WT_Amp. Notably, NCI-H596 cells express only the mutated allele (Fig S1A). Strikingly, in the native state, METΔ14 was not phosphorylated (active) but depended on the presence of HGF, similar to cells expressing wt MET (Fig 1C). By contrast, cells harbouring *MET* gene amplification, either wt or Δ14, showed constitutive activation, as expected (for review, see Hong et al [2021]). These surprising results encouraged us to study the HGF-dependent activation of METΔ14 further. Immunoblot (Fig 1C) and immunofluorescence (Fig S1B) experiments confirmed the dependence of METΔ14 to HGF. We also noticed that HGF stimulation resulted in more sustained METΔ14 activation than wt MET. Moreover, compared with the levels of MET wt protein decaying over time after HGF stimulation as reported before (Vigna et al, 1999; Kong-Beltran et al, 2006), METΔ14 resisted this down-regulation (Fig 1D), suggesting a longer half-life, possibly because of the loss of Y1003, responsible of c-CBL binding and MET degradation (Peschard et al, 2001). Dose–response experiments elicited that Δ14 was activated in the presence of lower HGF concentrations than wt (Fig 1E). These observations established altered kinetics and HGF affinity of METΔ14 compared with its wt counterpart.

To investigate the effect of long-lasting activation of METΔ14, we analysed the phosphorylation status of known downstream signal transducers such as ERK1/2 (for the RAS pathway), AKT (for the PI3K pathway), and STAT3 (Fig 2A and B) (Ponzetto et al, 1993, 1994; Fixman et al, 1996; Boccaccio et al, 1998). Intriguingly, we found that HGF-stimulated METΔ14 activation led to a robust AKT

phosphorylation, a less-pronounced ERK1/2 phosphorylation and undetectable STAT3 activation compared with wt MET (Figs 2A and B and S2A). To rule out the possibility that the strong AKT activation, the weak of ERK1/2 phosphorylation, and the less-pronounced STAT3 phosphorylation were cell line-specific features—that is, unrelated to exon14 deletion—we have stimulated cancer cells expressing either wt, Δ14, amplified wt, or amplified METΔ14 with HGF (Fig 2C). The cells expressing wt and METΔ14 displayed similar MET transcript levels (Fig S2B), whereas *MET* gene amplification status of wt or Δ14 were comparable (Fig S2C). Strikingly, AKT phosphorylation was restricted to cells expressing METΔ14 regardless of their amplified status. On the other hand, cells expressing the wt copy of *MET*—and not *MET*Δ14 displayed STAT3 and ERK1/2 (MAPK) activation. Moreover, transfection of TOV112D cells—known not to express endogenous MET—with either wt MET-Flag or METΔ14-flag vectors further confirmed the Δ14-specific activation of the AKT pathway (Fig 2D). These results suggest that METΔ14 induces a robust and selective AKT activation at the cost of STAT3 and ERK1/2 phosphorylation. The MET-dependence of AKT phosphorylation in METΔ14-amplified cells was further demonstrated by hampering activation of AKT in HGF-stimulated cells treated with specific MET inhibitors (Fig S2D). AKT activation is specific to METΔ14 and cannot be explained by an overactive receptor because it has not been observed in wt MET–amplified cells. The facilitated interaction between p85 and METΔ14 was previously reported and can justify the strong AKT phosphorylation in Δ14 cells treated with HGF (Lee & Yamada, 1995). Therefore, HGF-activated METΔ14 acts as a conditional gain-of-function mutation, with the ensuing AKT activation potentially giving rise to a selective survival advantage to cancer cells expressing METΔ14.

To deepen the analysis of pathways influenced by HGF stimulation, we performed transcriptomic analysis in cells expressing either wt MET, amplified wt MET, or METΔ14, stimulated for the indicated times with HGF. Strikingly, unsupervised analysis of the transcriptional response to HGF induced by METΔ14 formed a gene cluster distinct from those produced by either naïve or amplified wt MET (Fig 2E). The heatmap of the 500 most differentially expressed genes after HGF stimulation displayed a distinctive transcriptomic profile in METΔ14 (Fig 2F). Gene set enrichment analysis was then performed to compare hallmark gene sets disturbed by HGF. Notably, six pathways were exclusively enriched in cells expressing METΔ14: myogenesis (epithelial–mesenchymal transition [EMT]–like pathway), MYC targets V1 and V2, apoptosis, coagulation, and apical junction (Fig 2G). To exclude the possibility of a cell line-rather than METΔ14-specific effect, qRT-PCR was performed in three genetically diverse *MET* wt cells (based on the CCLE dataset [Ghandi et al, 2019]). Regardless of genetic background, the studied MET wt cells displayed a similar pattern of transcriptional activation for the selected genes, including EMT, mitotic spindle, and apoptosis upon HGF activation, which was distinct from that observed in METΔ14 cells. Specifically, HGF induced a substantial down-regulation of apoptosis-related genes in the latter. EMT genes were more substantially affected (i.e., mesenchymal markers up-regulated, epithelial markers downregulated); mitotic spindle-associated genes were not impaired (Fig S2E).

Collectively, these results suggest that in METΔ14-expressing cells, the PI3K/AKT pathway is selectively activated upon HGF

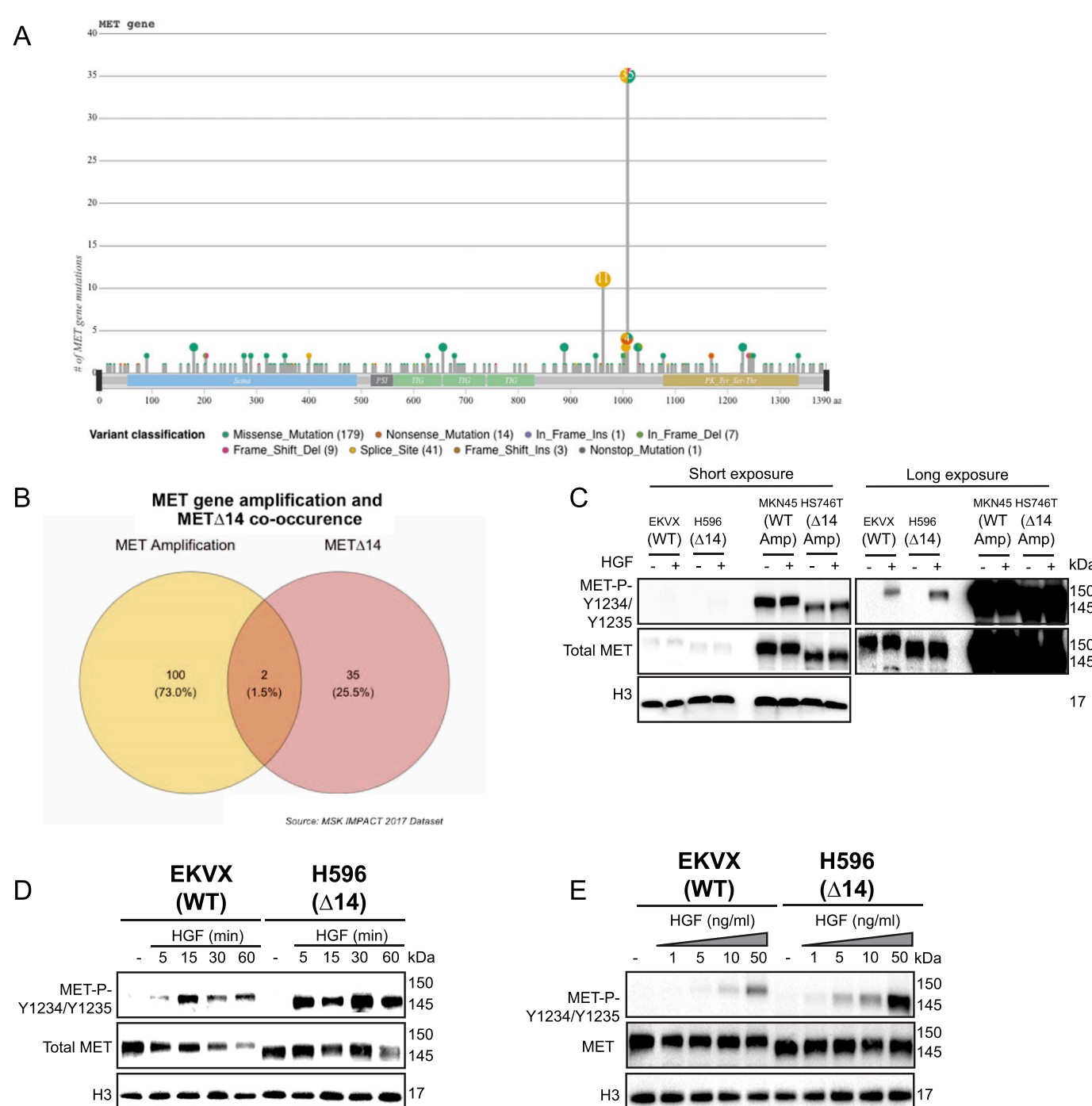

**Figure 1. METΔ14 activation requires hepatocyte growth factor (HGF).**
**(A)** Number of mutations in the *MET* gene based on the MSK-IMPACT dataset (N > 10,000 patients) (Zehir et al, 2017). **(B)** Venn diagram showing cooccurrence of *MET* amplification and METΔ14 in patients published in MSK-IMPACT dataset (Zehir et al, 2017). **(C)** Immunoblot experiments using protein extracts from cell lines harbouring indicated MET alterations, cultivated in serum-free medium ± HGF. **(D)** Kinetics of activation of wt MET and METΔ14 after the stated time of HGF treatment, assessed by immunoblotting. **(E)** Dose–response experiments to indicated concentrations of HGF in wt and METΔ14 cells. H3 was used as a loading control. Immunoblot pictures are representative of three independent experiments.
Source data are available for this figure.

stimulation. Accordingly, the transcriptomic profile altered by METΔ14 supports an HGF-dependent inhibition of apoptosis and enhanced survival and migration capabilities.

We have then directly tested the aforementioned assumptions. First, we performed a viability assay in the absence of serum and showed that METΔ14—and not wt MET—induced serum-independent

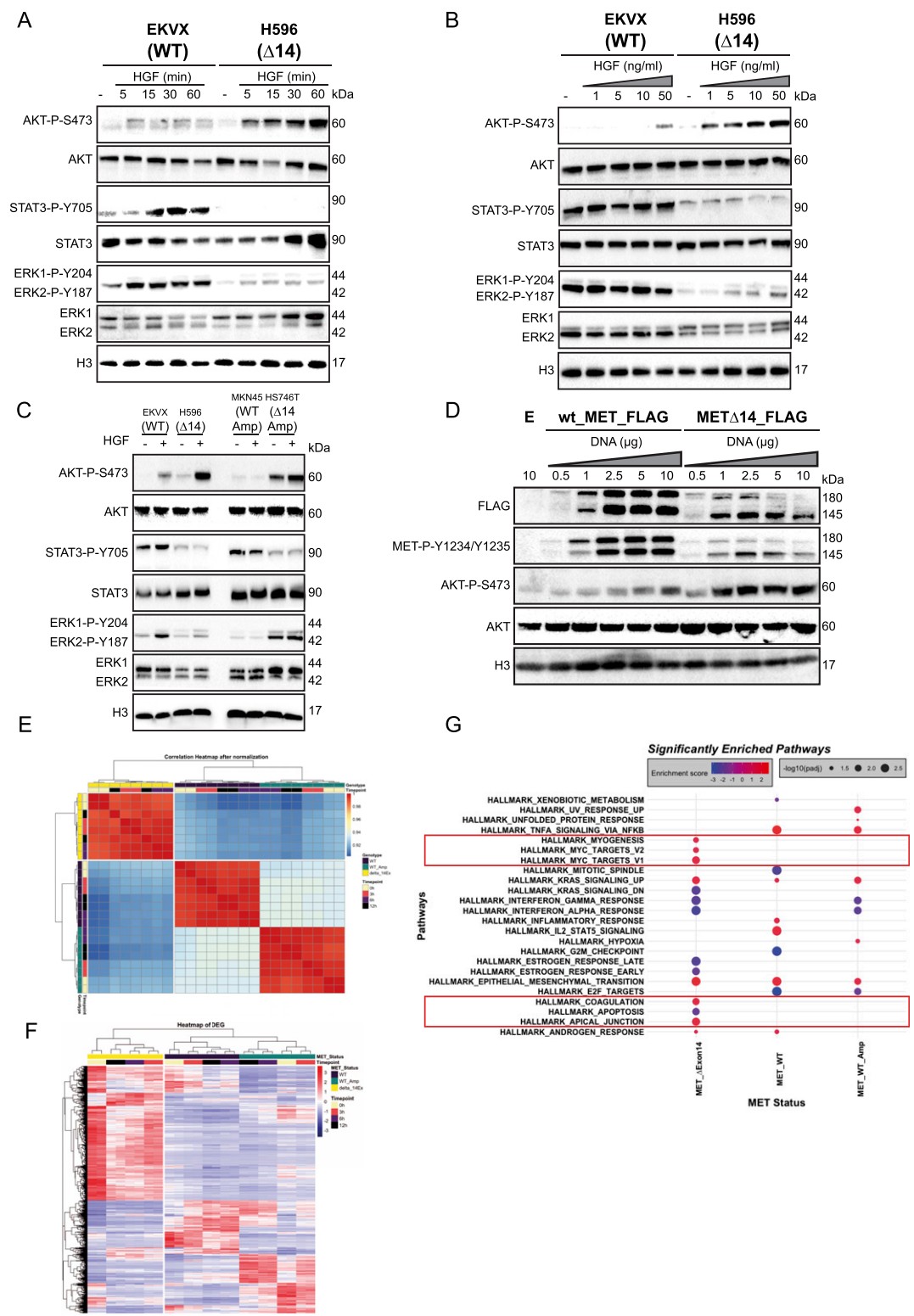

**Figure 2. Hepatocyte growth factor (HGF)-induced METΔ14 activation displays singular pro-oncogenic features.**
**(A, B)** Immunoblot exposing the phosphorylation status of known primary downstream signal transducer associated with HGF/MET pathway. **(A, B)** Cells expressing either wt or METΔ14 were stimulated with 50 ng/ml of HGF during indicated times (A) or with indicated concentrations of HGF during 15 min (B). **(C)** Activation status of MET downstream transducers was assessed by immunoblotting in cells expressing wt, METΔ14, amplified wt MET, or amplified METΔ14 treated with ± HGF. H3 was used as a loading control. Immunoblot pictures are representative of three independent experiments. **(D)** TOV112D cells were transfected with either pCDNA-3xFlag (empty vector, E), pCDNA-wt_MET-3xFlag, or pCDNA-METΔ14-3xFlag. Immunoblots were performed 48 h post-transfection. **(E)** Cells were treated with HGF for 0, 3, 6, or 12 h. Experiments

**Life Science Alliance**

growth (Fig 3A). Notably, the viability was compromised by the specific MET inhibitors tepotinib (Paik et al, 2020) and JNJ-38877605 (De Bacco et al, 2011). Second, we have treated cells expressing either wt or METΔ14 with etoposide, a commonly used genotoxic agent in lung cancer patients (Slevin et al, 1989). In the presence of HGF, METΔ14 cells were more effectively protected from etoposide-induced apoptosis than WT cells (Figs 3B and S3A). Third, to test how METΔ14 may promote cellular migration, we have measured wound healing induced by HGF (Fig 3C and Videos 1–Videos 16 online). Strikingly, METΔ14 completely covered the wound after 30 h whereas only half of the wound was covered in wt cells. In addition, faster wound healing was observed in METΔ14-amplified cells compared with MET_wt-amplified cells. In Δ14-amplified cells, the wound was healed in a ligand-independent manner. By contrast, cells expressing wt MET (amplified or not) never wholly covered the wound. In all cell lines studied, specific MET inhibitors abolished the process. These results suggest that faster wound healing is a particular feature conferred by METΔ14.

Increased viability and faster wound healing may result from METΔ14-induced cell proliferation. To explore a potential METΔ14-induced cell proliferation, cell cycle progression was assessed by EdU/PI labelling in METΔ14 and wt cells in concert with increasing concentrations of HGF (Fig S3B). In line with previous reports (Anastasi et al, 1997), HGF induced a modest concentration-dependent increase in the number of cells in the S phase in wt cells. Surprisingly, HGF did not increase the percentage of cells in the S phase in METΔ14. We conclude from these observations that the potent HGF-dependent wound healing response and the increased viability conferred by the HGF/METΔ14 axis are due to enhanced cell migration and survival rather than increased proliferation.

Thus, METΔ14 seems to act as a ligand-dependent gain-of-function enhancing cancer cell survival and migration capabilities. When activated by HGF released by the tumour microenvironment (e.g., by cancer-associated fibroblasts), METΔ14 activates the PI3K/AKT pathway without inducing STAT3 (differentiation) or MAPK (proliferation) phosphorylation. Thus, cancer cells increase their probability of survival within the tumour pseudo-organ, which possesses a myriad of unfavourable conditions, including shortage of nutrients, hypoxia, radiotherapy, and chemotherapy. We hypothesize that an enhanced migratory potential conferred by the METΔ14/PI3K/AKT axis allows cells to escape this hostile environment and form distal metastasis. Consequently, cells harbouring METΔ14 may choose to "fly" instead of "fight" locally by proliferation. Besides, migrating instead of dividing may constitute a shrewd strategy to escape an adverse local micromilieu for cell survival because most cancer drugs are designed to destroy actively dividing cells.

METΔ14 appears to be central to tumour cell survival. Hence in-depth understanding of the advantages it provides to cancer cells may be therapeutically exploited to target their vulnerability. However, the response to MET inhibitors is observed only in a fraction of Δ14 patients. Our present results suggest that the

resistance to targeted therapy may result from the absence of HGF in the tumour microenvironment or from activating mutations in the PI3K/AKT axis. Interestingly, the latter have been reported to coexist with MET exon 14 skipping in 14.2% of lung cancer patients (Jamme et al, 2020; Rotow et al, 2020).

Overall, our findings enlighten the mechanism of HGF-dependent METΔ14 activation and show that conditions other than mutations of the splicing site of exon 14 should be considered. Further preclinical and clinical work, including endogenous HGF levels and the mutational status PI3K/AKT axis, is required to justify the stratification of patients for clinical trials.

## Materials and Methods

### Reagents

Tepotinib (S7067), JNJ-38877605 (S1114), and etoposide (S1225) were purchased from Selleckchem. Recombinant HGF (294-HGN-025/CF) was bought from R&D Systems.

### Plasmids

3XFlag Tag was inserted into the pCDNA3.1+ vector between the HindIII and BamHI restriction sites. wt_MET or METΔ14 cDNA were inserted into pCDNA-3XFlag vector between NheI and HindIII restriction sites. Plasmids used in this study are available on Addgene with Addgene IDs 182494, 182495, and 182496.

### Cell culture

NCI-H596, HS746T, and TOV112D cells were purchased from ATCC. EKVX and MKN45 were obtained from the Cell Culture facility of Istituto Fondazione di Oncologia Molecolare (IFOM). Cells were cultured according to manufacturer's instructions and regularly verified for the absence of mycoplasma contamination. Experiments were performed before passage number 20. For HGF treatments, cells were serum-deprived for 24 h and treated with indicated time and concentrations of HGF. TOV112D cells were transfected with indicated quantity of DNA in 6-cm-diameter dishes using Lipofectamine2000 reagent (Cat. no. 11668019; Thermo Fisher Scientific) according to the manufacturer's instructions.

### MET alterations

Genomic DNA was extracted from cells using phenol/chloroform extraction. DNA was treated with RNase, and PCR reaction was performed using primers flanking the exon 14 of the *MET* gene (forward: 5'-GTCGTCGATTCTTGTGTGCTG-3'; reverse: 5'-GGGCTTCAA-CAGGTAAAAAATG-3') with the Phusion High-Fidelity DNA Polymerase (M0530; New England Biolabs). The PCR product was sequenced

were performed in biological duplicates. After sequencing, differential gene expression analyses were conducted using the R package DESeq2 version 1.32.0 with the formula ~Cell_line + time_of_HGF_treatment. The correlation heatmap was plotted using the pheatmap package. **(F)** Heatmap of top 500 differentially expressed genes after HGF treatment. **(G)** Gene set enrichment analyses on the genes whose expression values are significantly affected by HGF treatment.

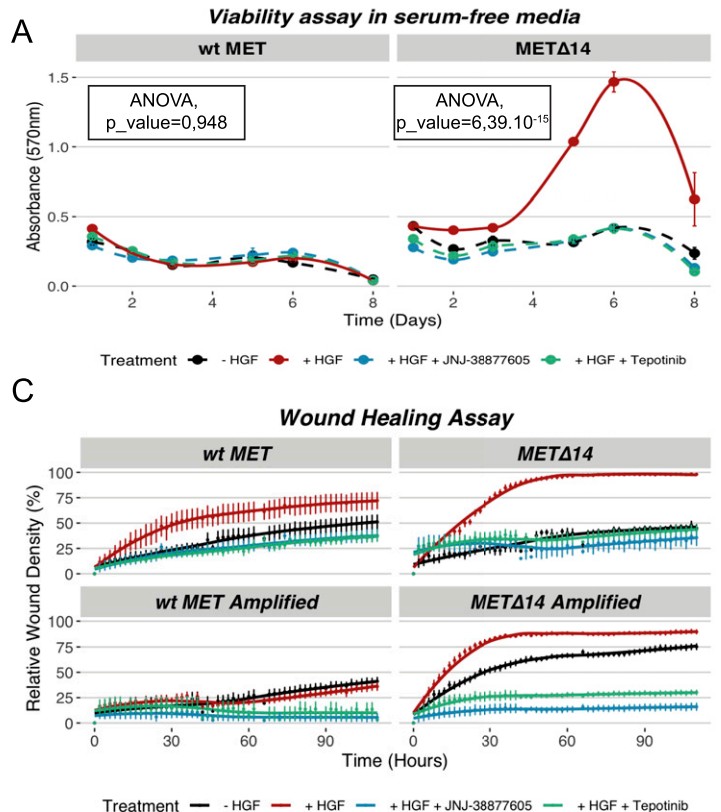

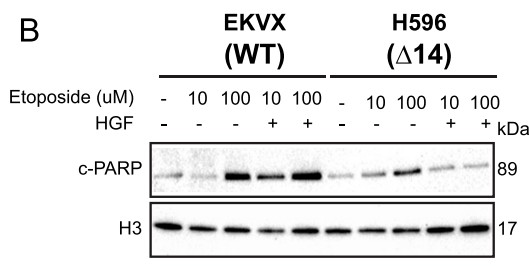

**Figure 3.** **Hepatocyte growth factor (HGF)–induced METΔ14 activation enhances viability, confers resistance to apoptosis, and increases the migration potential of cancer cells.**
**(A)** MTT viability assay in cells expressing wt MET or METΔ14 ± HGF ± MET inhibitors for the indicated time. Results are presented as absorbance values at 570 nm ± SEM (N = 6/condition). **(B)** Immunoblot to visualise c-PARP1 in cells treated with ± HGF ± etoposide for 12 h. H3 was used as a loading control. **(C)** Quantification of the relative wound density in cells expressing wt MET, METΔ14, wt MET–amplified, or METΔ14-amplified. Results are the spatial cell density in the wound area relative to the spatial cell density outside the wound area at every time point ± SEM (N = 6/condition).

by Sanger sequencing to assess *MET* mutational status. Besides, *MET* amplification status was evaluated using the MET Taqman probes (Hs04957390_cn and Hs05011082_cn). TERT (Hs02088500_cn) and RNAseP (Hs05182199_cn) genes were used as internal control for the assay.

### RNA extraction

Total RNA was extracted using QIAGEN RNeasy Mini Kit (Cat. no. 74106). For RNAseq experiments, quality assessment and library preparations were carried out as previously described (Nosi et al, 2021). Reads were trimmed to remove adapters sequences and mapped using STAR on ENSEMBL hg38 human genome assembly.

### Immunoblotting

Cells were lysed in Laemmli buffer as previously described (Modica et al, 2021). Proteins were resolved in pre-casted polyacrylamide gels (Thermo Fisher Scientific) and blotted against antibodies listed in Table S1.

### Immunofluorescence

Cells were plated in chamber slides (Thermo Fisher Scientific, Nunc Lab-Tek II Chamber Slide System, 154461 PK). Immunostainings were performed as previously described (Czibik et al, 2021). Antibodies are listed in Table S1. Images were acquired with the Leica TCS SP5 microscope.

### Viability assay

Cells were plated in 96-well plates at a density of 5,000 cells/well. After 24-h starvation, the cells were treated with ± HGF ± indicated MET inhibitors. Cell viability was assessed using CellTiter 96 Non-Radioactive Cell Proliferation Assay (G4100; Promega) every day for 80 d. The culture medium was renewed every 3 d.

### Wound healing assay

Cells were plated in 96-well plates at a density of 40,000 cells/well. After 24-h starvation, the cells were treated with ± HGF ± indicated MET inhibitors. After the wound generation, the relative wound

density was measured by Incucyte (Sartorius) according to the manufacturer's instructions.

### Apoptosis assay

Cells were treated for 18 h with 10 $\mu$M etoposide ± HGF. Apoptosis was assessed by immunoblotting using antibodies against cleaved PARP.

### Code availability

Data were analysed using R version 4.1.2. Codes are available at: https://github.com/Altintas-D/MET-14-promotes-a-ligand-dependent-AKT-driven-invasive-growth.

### qRT-PCR

1 $\mu$g of total RNA was reverse transcribed using Superscript III Reverse Transcriptase (Cat. no. 18080; Thermo Fisher Scientific) and relative gene expression levels were assessed using Taqman probes (Table S2). TBP and 18S were used as normalisation genes.

### Apoptosis assay

Cells were treated for 18 h with 10 $\mu$M etoposide ± HGF. Apoptosis was assessed by flow cytometry using the Violet Ratiometric Membrane Asymmetry Probe/Dead Cell Apoptosis Kit (Cat. no. A35137; Thermo Fisher Scientific).

### Cell cycle analysis

Cell proliferation was assessed by measuring EdU incorporation by flow cytometry. Briefly, cells were treated with indicated concentrations of HGF. After 2 h of EdU treatment, cells were pelleted and fixed, labelled with anti-EdU-Alexa Fluor647 and propidium iodide (DNA) using the Click-iT EdU Flow Cytometry Assay kit (C10424; Thermo Fisher Scientific).

## Data Availability

The RNAseq data from this publication have been deposited to Gene Expression Omnibus database (https://www.ncbi.nlm.nih.gov/geo/) and assigned the identifier GSE194382.

## Supplementary Information

## Acknowledgements

This project was funded by the AIRC-5x1000 (number 21052) and AIRC-19-IG (number 23820) grants. The authors thank the Istituto Fondazione di Oncologia Molecolare Imaging Unit, Cell Biology Unit, COGENTECH qPCR facility, COGENTECH DNA Sequencing facility, and COGENTECH Flow Cytometry facility for their precious technical contributions to experiments.

## Author Contributions

M Cerqua: conceptualization and methodology.
O Botti: methodology.
M Arigoni: methodology.
N Gioelli: methodology.
G Serini: methodology.
R Calogero: data curation.
C Boccaccio: conceptualization, funding acquisition, and writing—original draft, review, and editing.
PM Comoglio: conceptualization, funding acquisition, and writing—original draft, review, and editing.
DM Altintas: formal analysis, visualization, methodology, and writing—original draft, review, and editing.

### Conflict of Interest Statement

PM Comoglio acts as a Vertical Bio AG consultant. Other authors declare no competing financial interests.

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
