## [Reviewer comments · Life Science Alliance]

Life Science Alliance

MET Δ 14 promotes a ligand-dependent, AKT-driven invasive growth.

Marina Cerqua, Orsola Botti, Maddalena Arigoni, Noemi Gioelli, Guido Serini, Raffaele Calogero, Carla Boccaccio, Paolo Comoglio, and Dogus Altintas

DOI: <https://doi.org/10.26508/lsa.202201409>

Corresponding author(s): Paolo Comoglio, FIRC Institute of Molecular Oncology

Review Timeline:

Submission Date:	2022-02-15
Editorial Decision:	2022-04-12
Revision Received:	2022-04-22
Editorial Decision:	2022-05-12
Revision Received:	2022-05-13
Accepted:	2022-05-17

Transaction Report:

April 12, 2022

Re: Life Science Alliance manuscript #LSA-2022-01409-T

Prof. Paolo M. Comoglio
Fondazione Istituto FIRC di Oncologia Molecolare
Via Adamello 16
Milano, Lombardia I-10060 Candiolo (Torino)
Italy

Dear Dr. Comoglio,

Thank you for submitting your manuscript entitled "MET Δ 14 promotes a ligand-dependent, AKT-driven invasive growth." to Life Science Alliance. The manuscript was assessed and the Reviewer comments are appended to this letter. We invite you to submit a revised manuscript addressing the Reviewer comments.

The typical timeframe for revisions is three months. Please note that papers are generally considered through only one revision cycle, so strong support from the referee on the revised version is needed for acceptance.

When submitting the revision, please include a letter addressing the reviewer's comments point by point.

Thank you for this interesting contribution to Life Science Alliance. We are looking forward to receiving your revised manuscript.

Sincerely,

B. MANUSCRIPT ORGANIZATION AND FORMATTING:

Reviewer #2 (Comments to the Authors (Required)):

Cerqua et al. have investigated the role of mutations in the MET receptor that affect splicing of exon 14 (METdelta14) and result in in frame deletion of the juxtamembrane (JM) domain. Experiments are well-performed and figures nicely presented. However, the manuscript excludes many previous studies on this topic and most experiments are poorly controlled due to comparisons between entirely different cell lines.

- No mention is made in the manuscript of the original reports that METdelta14 is mechanistically distinct from the WT receptor (PMIDs: 7518457 and 17079873). These should be cited and their data discussed.
- The concept WT MET and METdelta14 activate distinct kinase networks has been previously reported in naturally occurring structural variants in mice (PMID: 7822270). Again, this prior literature is not cited or discussed in the present manuscript. The JM domain deletion led to increased association of the receptor with p85 PI3K, which would explain the increased Akt activation seen in the current manuscript.
- The first manuscript that identified a METdelta14 variant in a human cancer cell line might also be cited here, particularly as those authors also identified that the mutations lead to increased cell motility and altered signalling downstream of MET (PMID: 14559814).
- The role of MET protein stability which is thought to be crucial to the properties of METdelta14 (since the Y1003 c-Cbl binding site is located within the juxtamembrane region) is not addressed in the manuscript.
- In figures, the cell lines should be referred to by their names rather than as WT and delta14 and this misleadingly suggests that MET status is the only difference between these cells when they are in fact of a completely different genetic background (and even cancer type, since H596 cells are lung adenocarcinoma cells, HS746T and MKN45 cells are gastric adenocarcinoma cells, and EKVX are lung adenocarcinoma cells). In both the text and figure legends, all vague references to "cells" (e.g. lines 83, 128, 160 etc.) should be clarified with the specific name of the cell line used in experiments.
- The importance of cell line to cell line variability is evident in Fig 3B where the cell lines have different susceptibility to etoposide in the absence of HGF. It may be METdelta14 that confers a protective effect but there are many, many other differences between the WT and METdelta14 cell lines than simply MET status.
- Line 168: "independent of the cell line studied" - only one of the cell lines studies is METdelta14 amplified so how did the authors reach this statement? The Figure 3C figure legend should state what the error bars in this experiment represent.
- Throughout, figure legends should state the number of independent experiments each panel is representative of.
- Line 194: "allows cells to escape this hostile environment" - this sentence should be rephrased as a hypothesis as the data presented do not support this conclusion.
- Further pre-clinical studies are justified but in my opinion the authors suggestion that "endogenous HGF and the mutational status of the PI3K/AKT axis should be taken into serious consideration to select patients for clinical trials" is overly strong and not supported by the data presented here. The authors do not present in vivo or patient data to support their cell line work. How would such stratification be performed based on current data?

Minor/typos:

- There is a typo in Supplementary Figure 2 (Donnor).

Dear Editors, dear Reviewers

Thank you for reviewing our manuscript entitled “MET Δ 14 promotes a ligand-independent, AKT-driven invasive growth”. Please find below our answers, in bold characters, point by point to Reviewer 2’s thoughtful comments.

Reviewer #2 (Comments to the Authors (Required)):

Cerqua et al. have investigated the role of mutations in the MET receptor that affect splicing of exon 14 (METdelta14) and result in in-frame deletion of the juxtamembrane (JM) domain. Experiments are well-performed and figures nicely presented. However, the manuscript excludes many previous studies on this topic and most experiments are poorly controlled due to comparisons between entirely different cell lines.

- No mention is made in the manuscript of the original reports that METdelta14 is mechanistically distinct from the WT receptor (PMIDs: 7518457 and 17079873). These should be cited and their data discussed.

We acknowledge the Reviewer for this remark. The articles are cited in the ‘Introduction’ part, line 33.

- The concept WT MET and METdelta14 activate distinct kinase networks has been previously reported in naturally occurring structural variants in mice (PMID: 7822270). Again, this prior literature is not cited or discussed in the present manuscript. The JM domain deletion led to increased association of the receptor with p85 PI3K, which would explain the increased Akt activation seen in the current manuscript.

As with the previous comment, we added the citation and compared our results with the earlier data in the ‘Results and Discussion’ session, line 130.

- The first manuscript that identified a METdelta14 variant in a human cancer cell line might also be cited here, particularly as those authors also identified that the mutations lead to increased cell motility and altered signalling downstream of MET (PMID: 14559814).

The reference is now in the ‘Results and Discussion’ section, line 80.

- The role of MET protein stability which is thought to be crucial to the properties of METdelta14 (since the Y1003 c-Cbl binding site is located within the juxtamembrane region) is not addressed in the manuscript.

Thank you for this comment. We added this important mechanism explaining the enhancement of MET Δ 14 stability and cited the manuscript of Peschard *et al.*, 2001, on lines 33 and 98.

- In figures, the cell lines should be referred to by their names rather than as WT and delta14 and this misleadingly suggests that MET status is the only difference between these cells when they are in fact of a completely different genetic background (and even cancer type, since H596 cells are lung adenosquamous carcinoma cells, HS746T and MKN45 cells are gastric adenocarcinoma cells, and EKVX are lung adenocarcinoma cells). In both the text and figure legends, all vague references to “cells” (e.g. lines 83, 128, 160 etc.) should be clarified with the specific name of the cell line used in experiments.

We have presented the MET status (WT or Δ 14) instead of the entire cell line names in the figures and the text for clarity. We are fully aware that the MET mutational profile is not

the only difference between these cell lines. We performed transfection experiments on MET-negative TOV112D cells (Figure 2D). AKT phosphorylation is significantly enhanced when the cells are transfected with a plasmid encoding MET Δ 14 than those transfected with MET_wt. These results strengthened our conclusion that MET Δ 14 induces an HGF-dependent singular pathway, with AKT being the central transducer. However, we do not want to generate misunderstandings. A sentence stating the cell lines was added in the 'Results and Discussion' session (lines 79 to 84).

- The importance of cell line to cell line variability is evident in Fig 3B where the cell lines have different susceptibility to etoposide in the absence of HGF. It may be MET Δ 14 that confers a protective effect but there are many, many other differences between the WT and MET Δ 14 cell lines than simply MET status.

We partially agree with this statement. The level of PARP cleavage is comparable between EK VX (MET_wt) and NCI-H596 cells (MET Δ 14). The decline in HGF-dependent PARP cleavage, and thus apoptosis, has been observed only in Δ 14 cells (see the Immunoblot Caption from Figure 3B).

Moreover, we have performed RNAseq analyses in cells treated with +/- HGF. Since this manuscript is a short communication, we have not included the whole analysis. However, we are pleased to share with you the following data. Gene set enrichment analyses were performed in MET_wt cells (lung adenocarcinoma cells), MET Δ 14 cells (lung adenosquamous carcinoma cells), and MET_wt-amplified cells (gastric adenocarcinoma cells). Only cells expressing MET Δ 14 showed enrichment of the apoptosis pathway, with a significant, HGF-dependent decline of the pro-apoptotic genes' expression (barplot in the right panel).

- Line 168: "independent of the cell line studied" - only one of the cell lines studies is MET Δ 14 amplified, so how did the authors reach this statement? The Figure 3C figure legend should state what the error bars in this experiment represent.

We have modified the sentence to increase clarity. We have amended the legend of Figure 3.

- Throughout, figure legends should state the number of independent experiments each panel is representative of.

We apologize for this; the number of biological replicates is now present in the figure legends.

- Line 194: "allows cells to escape this hostile environment" - this sentence should be rephrased as a hypothesis as the data presented do not support this conclusion.

The sentence has been rephrased (line 198 of the amended manuscript).

- Further pre-clinical studies are justified but in my opinion the authors suggestion that “endogenous HGF and the mutational status of the PI3K/AKT axis should be taken into serious consideration to select patients for clinical trials” is overly strong and not supported by the data presented here. The authors do not present in vivo or patient data to support their cell line work. How would such stratification be performed based on current data?

We rewrote the final sentence to milden the conclusion about clinical trials (lines 217 to 219).

Minor/typos:

- There is a typo in Supplementary Figure 2 (Donnor).

Thank you, it is corrected in the current version.

We thank the Reviewer for her/his wise advice. We hope that the manuscript fulfils all the questions raised and is ready for publication.

Sincerely,

PM Comoglio

May 12, 2022

RE: Life Science Alliance Manuscript #LSA-2022-01409-TR

Prof. Paolo M. Comoglio
FIRC Institute of Molecular Oncology
Via Adamello 16
Milano, Lombardia 20139
Italy

Dear Dr. Comoglio,

Thank you for submitting your revised manuscript entitled "METΔ14 promotes a ligand-dependent, AKT-driven invasive growth.". We would be happy to publish your paper in Life Science Alliance pending final revisions necessary to meet our formatting guidelines.

- please address the Reviewer's final comments
- please add ORCID ID for both corresponding authors; you should have received instructions on how to do so
- please consult our manuscript preparation guidelines <https://www.life-science-alliance.org/manuscript-prep> and make sure your manuscript sections are in the correct order; please note the Results & Discussion sections are 2 separate sections
- please add your supplementary figure legends, video legends, and table legends to the main manuscript, directly under the main figure legends
- please add a callout for figure S2C to your main manuscript text
- please add a separate Data Availability Statement to indicate the accession information for the RNA-seq data

Figures:

- please provide the source data for Figure 1D
- In Figure 2C, is the space in the blots due to an empty lane, or spliced blots? If blots are spliced, please indicate.

A. FINAL FILES:

B. MANUSCRIPT ORGANIZATION AND FORMATTING:

Sincerely,

Reviewer #2 (Comments to the Authors (Required)):

The authors have addressed the majority of my comments and improved the manuscript during this round of revisions. My only remaining concern is the labelling of the cell lines in results text and figures, where I still feel that labelling cell lines in the format EKVX (WT), H596 (delta14) etc. would prevent the reader from inferring that WT and delta14 are isogenic lines if they hadn't read the manuscript in detail. Otherwise, I now believe the manuscript is suitable for publication.

Dear Editors, dear Reviewers

Thank you for judging our manuscript entitled “METΔ14 promotes a ligand-independent, AKT-driven invasive growth” suitable for publication. Please find below our point-by-point answers to the comments (bold character).

Dear Dr. Comoglio,

Thank you for submitting your revised manuscript entitled "METΔ14 promotes a ligand-dependent, AKT-driven invasive growth.". We would be happy to publish your paper in Life Science Alliance pending final revisions necessary to meet our formatting guidelines.

- please address the Reviewer's final comments
- please add ORCID ID for both corresponding authors; you should have received instructions on how to do so

ORCID IDs are now linked for the corresponding authors.

-please consult our manuscript preparation guidelines <https://www.life-science-alliance.org/manuscript-prep> and make sure your manuscript sections are in the correct order; please note the Results & Discussion sections are 2 separate sections

According to manuscript preparation guidelines, - “For shorter articles, the Results and Discussion sections can be combined.” – we made a single section referred to as ‘Results and discussion’.

-please add your supplementary figure legends, video legends, and table legends to the main manuscript, directly under the main figure legends

The legends are included in the appropriate order. Thank you.

-please add a callout for figure S2C to your main manuscript text

We are sorry for this mistake. The issue was corrected.

-please add a separate Data Availability Statement to indicate the accession information for the RNA-seq data

The statement is after the ‘Material of Methods’ section of the amended manuscript

Figures:

-please provide the source data for Figure 1D

Source data are uploaded as TIFF Files.

-In Figure 2C, is the space in the blots due to an empty lane, or spliced blots? If blots are spliced, please indicate.

In Figure 2C, the space is due to an empty line. That is to separate the MET amplified cells from the others. Indeed, we wanted to show a clear MET western blot for all the cell lines on the same gel without being affected by band intensity in the amplified cells (Figure 1C). The western blots corresponding to transducers (Figure 2C) were loaded similarly to the MET western blot for clarity and coherence.

To upload the final version of your manuscript, please log in to your account: <https://lsa.msubmit.net/cgi-bin/main.plex>

A. FINAL FILES:

B. MANUSCRIPT ORGANIZATION AND FORMATTING:

Sincerely,

Reviewer #2 (Comments to the Authors (Required)):

The authors have addressed the majority of my comments and improved the manuscript during this round of revisions. My only remaining concern is the labelling of the cell lines in results text and figures, where I still feel

that labelling cell lines in the format EKVX (WT), H596 (delta14) etc. would prevent the reader from inferring that WT and delta14 are isogenic lines if they hadn't read the manuscript in detail. Otherwise, I now believe the manuscript is suitable for publication.

Cell line names are added directly into the figures and supplementary figures to avoid misunderstandings. We thank the Reviewer for his positive deliberations about our work.

We hope that the manuscript fulfills all the questions raised and is ready for publication.

Sincerely,

PM Comoglio

May 17, 2022

RE: Life Science Alliance Manuscript #LSA-2022-01409-TRR

Prof. Paolo M. Comoglio
FIRC Institute of Molecular Oncology
Via Adamello 16
Milano, Lombardia 20139
Italy

Dear Dr. Comoglio,

Thank you for submitting your Research Article entitled "MET Δ 14 promotes a ligand-dependent, AKT-driven invasive growth.". It is a pleasure to let you know that your manuscript is now accepted for publication in Life Science Alliance. Congratulations on this interesting work.

DISTRIBUTION OF MATERIALS:

Again, congratulations on a very nice paper. I hope you found the review process to be constructive and are pleased with how the manuscript was handled editorially. We look forward to future exciting submissions from your lab.

Sincerely,
